# Application of a Water Supply-Demand Balance Model to Set Priorities for Improvements in Water Supply Systems: A Case Study from the Koshi River Basin, Nepal

**DOI:** 10.3390/ijerph19031606

**Published:** 2022-01-30

**Authors:** Ran Zhu, Yiping Fang

**Affiliations:** 1Institute of Mountain Hazards and Environment, Chinese Academy of Sciences, Chengdu 610041, China; zhuran0520@foxmail.com; 2University of Chinese Academy of Sciences, Beijing 100049, China; 3College of Resources and Environment, University of Chinese Academy of Sciences, Beijing 100049, China; 4China-Pakistan Joint Research Center on Earth Sciences, CAS-HEC, Islamabad 45320, Pakistan

**Keywords:** water vulnerability, water supply-demand gap, priority setting, improvements, water supply systems, Koshi River basin

## Abstract

Water scarcity is one of the leading challenges for sustainable development in the context of climate change, particularly for agriculturally reliant countries. Inadequate water supplies tend to generate environmental and health issues. Improvements in water supply systems should give priority to the region with the most severe mismatch between water supply and demand. To set priorities for the improvement of water supply systems, this study proposed a water supply-demand balance model to quantify the water supply-demand gap in the Koshi River basin and compared it with the traditional water vulnerability model. The results suggested that (1) the water supply-demand balance model had good applicability to the Koshi River basin and was superior to traditional models in identifying the region with the most severe mismatch; (2) the shortage of agricultural water was much more serious than that of domestic water in the basin; (3) the largest supply-demand gap of domestic water was in Tarai and that of agricultural water was in the hill areas; and (4) Four districts, including Lalitpur, Mahottari, Makwanpur, and Solukhumbu, were found to be the most water-stressed regions and priority should be given to them. Based on these findings, the priority setting in the improvement of water supply systems and adaptation strategies for mitigating water stress from the perspectives of the government, communities, and households were presented. It helps design water supply systems that match heterogeneous demands and optimize systems operation. Targeted improvements in water supply systems can make limited funds available to benefit more residents.

## 1. Introduction

Water scarcity is becoming a fundamental challenge globally [1,2,3,4]. In the 2013 Global Risks report, water supply crises were identified as one of the greatest impacts and most likely challenges facing the planet, and these crises have greatly restricted economic development and worsened poverty [5,6]. There are multiple lines of evidence that most parts of the world, especially agricultural-dominant areas, have perceived the negative impacts of climate change on water supply, and the supply-demand contradiction will lead to more severe water shortages in the future [7]. Many residents in developing countries are disadvantaged concerning access to adequate domestic and agricultural water because of the severe mismatch between water supply and demand resulting from the effects of climate change, environmental degradation, the low performance of existing water supply systems, and improper management of these systems and their sustainability [8,9]. Human-induced climate change, with its negative impacts on the water supply in many less developed regions, has given rise to sustained interest in assessments of water stress [10]. Vulnerability studies have found that households living in mountainous areas always face the highest water stress due to diverse topography, agricultural-dominant production models, and poor infrastructure [11,12]. Furthermore, the water demand is growing with population growth and rapid urbanization, whereas water sources are drying up because of frequent drought disasters, deforestation, and environmental degradation. It is imperative for water supply and environment improvement to evaluate water stress, especially in less developed mountains with poor infrastructure.

Researchers developed the water vulnerability index (WVI) [13,14,15], which incorporates a wide range of natural and socioeconomic variables to assess water stress. The WVI is usually framed as an integration of a physical subindex (physical processes affecting water resources) and a social subindex (potential capital to adapt to changing water resources) [16]. Some researchers have defined water vulnerability as a more detailed component, including resource stress, ecosystem health, development pressure, and management capacity, for horizontal comparison [17]. To clarify the impacts of water vulnerability on a range of aspects of the economy and society, Sullivan [18] constructed the WVI composed of the supply-driven vulnerability of water systems and demand-driven vulnerability of water users. However, the vulnerability model cannot quantify the water supply-demand gap to identify the region with the most severe mismatch.

Previous research on supply and demand situations has shown that demand-side factors, such as rapid urbanization, economic development, and massive population growth, seem to have a stronger impact on the supply-demand balance of domestic water than do the supply-side drivers, represented by climate variability in megacities [19,20,21]. Conversely, the domestic water supply in rural regions with poor infrastructure is likely to be more sensitive and vulnerable to climate change. Research on the supply and demand situations of agricultural water concentrated mainly on the impacts of the ecological environment [22], climate change [23] and human activities [24], the patterns of agricultural irrigation water and cropland allocation [25], and strategies for the improvement of water supply systems, including rainwater harvesting (RWH) systems [26], water delivery for irrigation and water resources management [27]. The system dynamics model [19], hydrological model, and water evaluation and planning model [28] have been constructed to calculate the water supply-demand gap in previous studies, and all of these models have been applied to assess the supply-demand gap in cities. These models are not suitable for less developed mountainous areas with poor water supply facilities because they emphasize the importance of the total amount of water resources, ignoring the impacts of the lack of ability to exploit and deliver water resources on water supply capacity. In addition, the data regarding the water supply at different hydraulic levels of the scheme and water demand (e.g., cropping patterns) must be complete in these models, which makes it difficult to obtain data, especially in less developed mountains with incomplete data [29].

Based on previous studies on water vulnerability and supply and demand situations, this study proposed a water supply-demand balance model that separated supply factors from demand factors of vulnerability variables to quantify the supply-demand gap. It was designed for the assessment of water resource stress in less developed areas with poor water supply facilities rather than for cities, as in previous models. Domestic water is required for a wide range of activities, including drinking, cooking, and washing, and a sufficient water supply can contribute to poverty alleviation. In addition, agricultural development is a powerful tool to end extreme poverty and boost income in most less developed countries, and adequate quantities of agricultural water can help achieve the Sustainable Development Goals (SDGs) related to eliminating hunger. However, the impacts of the supply of urban ecological water and industrial water on residents’ livelihood security in these areas are not significant [30]. Thus, the supply-demand balance model was constructed for assessments of domestic and agricultural water stress. Finally, for further comparison, both the vulnerability model and the supply-demand balance model were applied to the Koshi River basin (KRB), where people are susceptible to the effects of climate change [30]. The aims of this paper were threefold: first, to construct a water supply-demand balance model for areas with poor water supply facilities and compare the merits and demerits of the vulnerability and supply-demand approaches; second, to identify the region with the most severe mismatch between water supply and demand; and third, to present the priority setting in the improvement of water supply systems and adaptation strategies for mitigating water stress from the perspectives of the government, communities, and households.

## 2. Methodology

### 2.1. Study Area: The Koshi River Basin of Nepal

We used the KRB as the study area. The Koshi River is an important transboundary river that originates on the Tibetan Plateau of China, passes through Nepal, and then crosses the floodplain areas of Bihar, India, eventually ending at the Ganges. It runs from north to south for a total length of 255 km, covering an area of 87,970 km^2^, approximately 33% in China, 45% in Nepal, and 22% in India, sustaining approximately 40 million people [31]. The elevation ranges from 8844 masl to approximately 21 masl from north to south [32]. This study focuses on the part of the Koshi River basin in Nepal, composed of 27 districts of eastern Nepal, covering an area of 39,500 km^2^, as shown in Figure 1.

Precipitation is influenced by both topography and monsoons in this region, and approximately 80% of the total annual precipitation occurs in the monsoon season. Uneven distribution of precipitation is likely to lead to seasonal drought, which threatens water security and food production. The ongoing change in climatic regime tends to change the water cycle along the KRB, altering the time, magnitude, and intensity of precipitation as well as affecting evaporation. This change could translate into wetter wet seasons and drier dry seasons, posing challenges to people who depend heavily on agriculture in the KRB [33].

The KRB has been at a low level of urbanization, and the agricultural-dominant production model results in high demand for agricultural water. Severe issues of chronic poverty and an unreliable water supply tend to aggravate vulnerability to climate change and widen the gap between the water supply and demand. The extremely rugged terrain and scattered settlements encourage rural residents to build and manage their water supply systems but also make the process more difficult [34]. Research has shown that only approximately half of the population has access to piped water in the KRB, and most local water user committees are unable to operate schemes sustainably due to financial difficulties, a low capacity to carry out maintenance, and the lack of accountability or transparency [35]. In addition, the irrigated area is only approximately half of the cultivated land, and it does not receive year-round irrigation, which has been regarded as one of the major constraints on agricultural development in Nepal [36,37].

According to the geographical structure, the 27 districts in the KRB have been divided into three geographical and agro-ecological regions, including the mountain region (5 districts), the hill region (14 districts), and the Tarai region (8 districts), in previous studies. There were significant differences in climate, elevation, terrain, population density, economic development, and livelihood activities among the three regions [38,39]. Therefore, the results of the water supply-demand assessment will be discussed along the three agro-ecological regions in this paper.

### 2.2. Three Agroecological Regions

The mountain districts fall under the lap of the high Himalayan range with severe climate conditions and difficult terrain with scattered settlements, and the elevation ranges from 2500 masl to 8848 masl [39,40]. This region is not suitable for agricultural production and is difficult to harness economically due to the hostile climate, infertile land, and poor physical infrastructure.

The range has an average elevation of 200 masl to 2500 masl in the hill districts, and this region is characterized by subtropical and temperate climates [40]. People are mostly engaged in terrace farming. Additionally, labor exports are a powerful tool to boost poverty alleviation in hills.

The Tarai is located in northern India and southern Nepal, and the elevation ranges from 21 masl to 200 masl [31,39]. It is densely populated with high agricultural productivity due to subdued topography, large areas of fertile land, and relatively better infrastructure development.

### 2.3. Frameworks for the Water Vulnerability and Supply-Demand Balance Index

The traditional vulnerability approach uses multilevel indices to calculate the overall WVI composite index, which is difficult to use for horizontal comparisons of different categories of influencing factors. Therefore, based on the WPI [41] and water vulnerability index [18], this paper proposed the WVI incorporating five components (environment, resource, access, use, and capacity) in consideration of the significant impacts of environmental and climatic change, access to water sources, coverage of water supply systems and capacity to cope with drought disasters on water stress. Environment denotes the ecosystem goods or services from aquatic habitats and the environmental integrity, which is related to water. Resources provide assessments of the number of water resources. The access component depicts access to adequate water and sanitation. Use indicates the ability to exploit and deliver water resources. The capacity component shows the ability to maintain livelihoods in times of water shortages.

Furthermore, this study separated supply-driven factors from demand-driven factors and constructed supply-demand balance models of domestic and agricultural water to calculate the gap between water supply and demand caused by multiple factors, including climate change, poor infrastructure, and long distances to water resources. The construction process of frameworks for the water vulnerability and supply-demand balance index is shown in Figure 2. From the vulnerability model to the supply-demand model, variables relevant to supply-driven vulnerability and demand-driven vulnerability were adjusted to those relevant to the water supply capacity (DSI and ASI) and the water demand level (DDI and ADI), respectively.

The process of water supply and demand in the KRB is represented in Figure 3, which can help us understand the water supply and demand situations. The WSDBI (water supply-demand balance index) incorporates the DI (water supply-demand balance index for domestic water) and the AI (water supply-demand balance index for agricultural water). In the framework for calculating the WSDBI, water supply capacity (supply-driven factors) is measured by both the total amount of water resources and the ability of water utilization in consideration of the lack of access to water in Nepal. The water demand level is determined by the demand for domestic water (used for drinking, washing, cooking, etc.) and agricultural water (used for irrigation, livestock drinking, etc.).

### 2.4. Calculating the WVI and WSDBI

#### 2.4.1. Calculating the WVI: A Composite Indices Method

To ensure the experimental results are of scientific quality, we adopted the balanced weighted average approach to model the vulnerability to water shortages based on the assumption that each component equally contributes to the WVI [39,42]. According to published research that is significant to the KRB, typical features of the study area, and data availability, 14 indicators were selected to compose the five components of the WVI [12,41]. The original variables need to be standardized first to eliminate the dimension. The data standardization approach (Equations (1) and (2)) is widely used to calculate a composite index [43,44]. After each indicator was standardized, Equation (3) was adopted to calculate the overall WVI composite index. The values of the five components all range from 0 to 0.2, and the higher the component value, the higher the WVI.

For the indicators with a positive correlation with the corresponding component of the WVI:(1)$$r_{i}=(\lambda_{i}-\lambda_{imin})/(\lambda_{imax}-\lambda_{imin})$$

For the indicators with a negative correlation with the corresponding component of the WVI:(2)$$r_{i}=(\lambda_{imax}-\lambda_{i})/(\lambda_{imax}-\lambda_{imin})$$
where $\lambda_{i}$ is the original indicator subcomponent; $\lambda_{imax}$ and $\lambda_{imin}$ are the maximum and minimum values, respectively; and $r_{i}$ refers to the standardized value of $\lambda_{i}$. This standardization makes for data with values ranging from zero to one to make variables comparable to each other.
(3)$$WVI=\frac{\omega_{e}E+\omega_{r}R+\omega_{a}A+\omega_{u}U+\omega_{c}C}{\omega_{e}+\omega_{r}+\omega_{a}+\omega_{u}+\omega_{c}}$$
where $\omega_{e}$, $\omega_{r}$, $\omega_{a}$, $\omega_{u}$, $\omega_{c}$ are the weights of the five components of the WVI [environment (E), resource (R), access (A), use (U) and capacity (C)], respectively.

#### 2.4.2. Calculating the WSDBI

The advantage of the supply-demand model is to quantify the gap between water supply and demand, so the supply-driven and demand-driven factors separated from the vulnerability model were adjusted to match the demands of the supply-demand model. Based on a review of the literature, typical features of the study area, and data availability [12,16,18,20,45,46,47,48,49,50], 7 indicators were selected to develop the DI, and 8 indicators were selected to construct the AI. The whole process consisted of four major steps: (A) All variables were first standardized, and the standardization approach was written as Equations (1) and (2). (B) The combination of the entropy weight method and expert scoring method was adopted to distribute the weight of indicators. (C) Equations (4) and (5) were used to calculate the DI and AI, respectively, as follows. The DSI, DDI, ASI, and ADI all range from 0 to 1. The higher the DSI/ASI and the lower the DDI/ADI, the higher the DI/AI. If the DI/AI is less than 1, supply is less than demand; if the DI/AI is equal to 1, water supply and demand are in balance; if the DI/AI is more than 1, demand is less than supply.
(4)$$DI=DSI/DDI=r_{DSI_{1}}/r_{DDI_{1}}=r_{\sum_{k_{DS}=1}^{m_{DS}}r_{k_{DS}}\omega_{k_{DS}}}/r_{\sum_{k_{DD}=1}^{m_{DD}}r_{k_{DD}}\omega_{k_{DD}}}$$
(5)$$AI=ASI/ADI=r_{ASI_{1}}/r_{ADI_{1}}=r_{\sum_{k_{AS}=1}^{m_{AS}}r_{k_{AS}}\omega_{k_{AS}}}/r_{\sum_{k_{AD}=1}^{m_{AD}}r_{k_{AD}}\omega_{k_{AD}}}$$
where *m* is the number of indicators and ω is the weight of the corresponding indicator. $k_{DS}$ and $k_{AS}$ are the indicators related to domestic water supply capacity and agricultural water supply capacity, respectively. $k_{DD}$ and $k_{AD}$ are the indicators related to the domestic water demand level and agricultural water demand level, respectively. DSI, DDI, ASI and ADI refer to the standardized values of $DSI_{1}$, $DDI_{1}$, $ASI_{1}$ and $ADI_{1}$, respectively. The data standardization approach is the same as that shown in Equations (1) and (2).

### 2.5. Data Sources

The meteorological data for this study mainly included the annual precipitation and coefficient of variation of precipitation from 1979–2020. These data from all stations within 27 districts were provided by the Department of Hydrology and Meteorology (DHM), Nepal. Since data for the annual water resources were unavailable, we applied specific discharge calculated by Sinha et al. [51] to estimate it, and the complete process was represented in Manandhar et al. [11]. In addition, the distance between houses and water sources was calculated by using the Nepal land cover provided by the International Centre for Integrated Mountain Development (ICIMOD). The socioeconomic data for other indicators were taken from published studies that are of significance for the development of Nepal. The data used in this paper are all from the latest datasets published by various government departments of Nepal (e.g., ICIMOD, Central Bureau of Statistics, and Department of Hydrology and Meteorology Nepal). They are authoritative and have been used in much research. The details are shown in Table 1.

## 3. Results

### 3.1. The WVI for the Koshi River Basin

#### 3.1.1. Comparing the WVI of the Environment, Resource, Access, Use, and Capacity Components

The results of the WVI and vulnerability of the five components are shown in Figure 4. Use showed the highest vulnerability (18.85), followed by access (13.89) and capacity (13.60), and the vulnerability of resource (6.86) and environment (6.30) was significantly lower than the others. The results suggested that the main water issues in the KRB were the lack of ability to exploit water resources and the low capacity to cope with water shortages rather than a shortage of water resources or the ability to conserve headwater. Specifically, the hills and a few districts of the mountain region had the highest access vulnerability. It was difficult for residents living in these mountain villages to exploit groundwater because of the rugged terrain. Water provided by water trucks was barely enough for domestic use, let alone irrigation. It is difficult to access locations and build water supply systems in the mountain region. The use component had the highest vulnerability among the five components. The improvement of water conservancy facilities is an effective means to transform groundwater and springs into domestic water and agricultural water. In addition, measures such as rationally levying water charges to promote further awareness of water savings would help improve water utilization. The spatial distribution of capacity vulnerability was similar to that of use vulnerability, showing a strong correlation between the two components. Therefore, strategies for reducing the vulnerability of one component may lead to a decrease in the vulnerability of both.

#### 3.1.2. The WVI for Three Agro-Ecological Regions

According to the results, higher water vulnerability was found in the hills (67.47) and mountain regions (65.09), mainly because both areas showed poorer irrigation infrastructure. In the hills and mountains, 66.71% and 65.26% of arable land did not receive irrigation, respectively, and inhabitants had to depend heavily on monsoon rainfall. In addition, the long distance to water sources and the rugged terrain made it difficult to deliver water. The lowest vulnerability was found in Tarai (42.05). However, approximately 77.01% of households lack access to proper sanitation in this region, which may put people at risk of disease. Thus, how to provide adequate, clean, and safe drinking water has become an urgent problem for this area. Furthermore, unlike other districts in the hills, lower water vulnerability was found in Kathmandu, the capital of Nepal, and its neighboring districts, mainly because nonfarm income boosted the regional capacity to deal with water shortages and further reduced its vulnerability.

### 3.2. The WSDBI for the Koshi River Basin

#### 3.2.1. The DI and AI at Different Scales

*The comparison of DI and AI in the Koshi River basin.* The WSDBI for domestic water (DI) in the KRB was 1.75, and it was 0.74 for agricultural water (AI), indicating that the supply-demand gap of agricultural water was larger than that of domestic water, mainly because the coverage of water supply facilities was larger than that of irrigation infrastructure. Based on previous studies, researchers have reported that it was difficult for lands above the level of the water sources to obtain access to irrigation water because the geographic structure of Nepal was mostly mountainous [57]. Moreover, most community-managed irrigation systems are gravity-based rather than storage type, which is likely to lead to shortages of agricultural water in the dry season [53,58]. These findings explain why people in this basin always suffer from shortages of irrigation water.

*Comparison of the DI and AI along three agro-ecological regions*. Comparing the DI in the three selected agro-ecological regions, the supply-demand gap was the largest in Tarai because of the high demand for domestic water. It was reported by the Central Bureau of Statistics that Tarai was the most densely populated area, with a population of 13.32 million in the three agro-ecological regions, which resulted in high demand for domestic water. The largest supply-demand gap of agricultural water was found in the hills, mainly due to insufficient irrigation infrastructure. In the Tarai districts, 62.52% of the agricultural area was equipped with irrigation infrastructure to provide water to crops, but in the hill regions, only 29.36% of the agricultural area was equipped with irrigation infrastructure, which greatly increased water stress, especially in the non-monsoon season.

*The comparison of the DI and AI at the district level.* The district-level comparison for the WSDBI and the two elements, water supply, and demand, are presented in Figure 5. The DI and AI are shown in Table 2. Specifically, in 18.52% of districts, the DI was more than 1, and in 74.07% of districts, the AI was less than 1, which meant that the domestic water for households in most districts was sufficient, but the agricultural water was in short supply. Therefore, decision-makers should prioritize the improvement of agricultural water supply systems, which is an urgent need in most districts. With respect to domestic water, lower DI was found in some districts, including Sunsari (Tarai), Solukhumbu (Mountains), Taplejung (Mountains), Lalitpur (Hills), and Bhaktapur (Hills). The scattered distribution of districts with low DI suggested that domestic water shortages were likely to be caused by a wide range of natural and socioeconomic factors, such as climate, topography, and infrastructure, rather than by a single factor. Lower AI was found in some districts, including Okhaldhunga (Hills), Kavrepalanchok (Hills), Panchthar (Hills), Ramechhap (Hills), and Makwanpur (Hills). The concentrated distribution of these districts indicated that poor irrigation infrastructure was likely to be the main cause of agricultural water shortages.

#### 3.2.2. The Combination of the DI and AI

The combination of the DI and AI is a powerful tool to identify the types of water issues in different districts or agro-ecological regions. Therefore, by comparing the DI and AI of each district with the median value of the region, the high-DI high-AI group (HH group), high-DI low-AI group (HL group), low-DI high-AI group (LH group), and low-DI low-AI group (LL group) were identified for further analysis. Based on the combination of the DI and AI, the spatial distribution of water issues is clearly reflected in Figure 6, which shows that most districts faced either a shortage of domestic water or a shortage of agricultural water, while a few districts faced risks from a shortage of both domestic and agricultural water or neither. Districts in the HH group, including Terhathum, Udayapur, Sankhuwasabha, and Sarlahi, were in the least water-stressed situation. The HL group consisted of 9 districts: Siraha, Sindhupalchok, Ramechhap, Panchthar, Okhaldhunga, Khotang, Kavrepalanchok, Dhankuta, and Bhojpur. They were in short supply of agricultural water rather than domestic water. Conversely, districts in the LH group, including Bara, Bhaktapur, Dhanusa, Dolakha, Kathmandu, Rautahat, Saptari, Sindhuli, Sunsari, and Taplejung, lacked domestic water rather than agricultural water. Four districts (Lalitpur, Mahottari, Makwanpur, and Solukhumbu) were in the LL group, and these districts were found to be the most water-stressed regions due to shortages of both domestic and agricultural water. As shown in Figure 6, districts in Tarai were more likely to have a large supply-demand gap of domestic water, while the large supply-demand gap of agricultural water was more likely to occur in the hilly regions.

## 4. Discussion

### 4.1. Comparison of the Water Vulnerability and Supply-Demand Balance Models

The water vulnerability model can be used for horizontal comparisons of different components’ vulnerability, and the result can be combined into an index that is quite clear. The results showed that the hill areas had the highest vulnerability, and that Tarai had the lowest vulnerability. However, if we do not use the supply-demand balance model, which separates supply factors from demand factors of vulnerability variables for further analysis, the water stress, and scarcity in the Tarai districts are likely to be ignored due to its low water vulnerability. The water supply-demand model can effectively quantify the supply-demand gap of domestic and agricultural water, and the results indicated that Tarai had the largest supply-demand gap of domestic water in the three agro-ecological regions and that the hill areas had the largest supply-demand gap of agricultural water. This finding has been supported by many water-related studies in Nepal. Khadka and Pathak [50] thought that flat to gentle slope areas in Tarai was a very good category for groundwater storage because of the subdued topography and the slow surface runoff allowed more time for rainwater to percolate, which was conducive to the agricultural water supply. Biswas [59] reported that irrigation developments in the hilly region were restricted by topographical conditions. Specifically, water from the large and medium-sized rivers that flow through the hills cannot be used because they cut deep through the region, meaning that water levels are at too low an elevation compared to the fields to be irrigated effectively. These findings further explain why the large supply-demand gap of agricultural water was more likely to occur in the hill regions rather than the Tarai. The large supply-demand gap of domestic water in Tarai was due to the insufficient water supply facilities that were used to support the increasing water demand caused by the growing population and rapid urbanization, which can be supported by the finding that rapid urbanization may aggravate the problem of water supply [18]. It is a common occurrence that women and children wait in long lines for their turn to collect limited water in urban areas. The high population density in these areas leads to a large demand for domestic water, and clustered houses are not conducive to rainwater collection.

### 4.2. Identifying the Region with the Most Severe Mismatch between Water Supply and Demand

The most water-stressed districts, including Lalitpur, Mahottari, Makwanpur, and Solukhumbu, were identified based on the combination of the DI and AI. People living in these regions were more likely to suffer from hunger and poverty due to the risks from the shortage of both domestic and agricultural water, which may increase the time and cost of water collection. Some studies have suggested that households without water supply facilities usually had to spend several hours a day fetching water from rivers [60] and that inhabitants had to pay for a higher cost of water than people who could afford piped water [18]. Land entitlement issues, lack of awareness, and high connection costs are major reasons for them remaining unconnected to the system. Funds for the construction of water supply systems should be prioritized in these districts. This finding is incredibly valuable because it highlights the region with the most severe mismatch between water supply and demand, which helps to attract the attention of the government, nongovernmental organizations (NGOs), and communities, and provides a reference for them to set priorities for the improvement of water supply systems.

### 4.3. Priority Setting in the Improvement of Water Supply Systems

During the period from 1993 to 2012, it was observed that the proportion of households with access to safe and secure water declined from 77.6% to 61.1% in Nepal [61]. The lack of access to clean and safe drinking water may lead to a variety of avoidable diseases, which in turn can impose economic costs on the poor [52]. Improving water supply systems is an important way to reduce poverty. Moreover, it can make visible the level of progress made in achieving SDG 6, which aims to ensure access to water and sanitation for all by 2030 [62]. Furthermore, high coverage of irrigation infrastructure can counteract the negative effects of long distances from the field to water sources. Access to an easily accessible, sustainable, affordable water supply is crucial for agricultural productivity. Despite the abundant water resources in Nepal, the shortage of irrigation water is a major challenge for the country due to insufficient irrigation infrastructure. Government-managed irrigation systems fail to consider the micro-issues related to the supply and demand of water. The priority setting in the improvement of water supply systems at different scales, which is favorable to designing water supply systems that match heterogeneous demands, is based on the water supply-demand gap, as shown in Figure 7. In the KRB, priority should be given to the improvement of agricultural water supply systems. In the three selected agro-ecological regions, the improvement of domestic water supply systems should give priority to Tarai, and improvements in agricultural water supply systems should give priority to the hills. In addition, in the 27 districts of the KRB, priority should be given to the LL group because it is the most water-stressed area due to shortages of both domestic and agricultural water.

From the perspectives of the government, communities, and households, adaptation strategies for the improvement of water supply systems were proposed, as shown in Figure 7. Mitigating water stress requires the coordination and cooperation of multiple stakeholders. The government plays a key role in policy formulation and implementation. Communities are the link between the government and inhabitants. They can provide technologies and services to residents to cope with droughts and raise money from the government, NGOs, or inhabitants to build water storage or irrigation systems. Livelihood diversity, collecting rainwater, and improving water resource utilization efficiency are strategies that households can adopt to reduce their water vulnerability.

### 4.4. Limitations of This Study

Due to the lack of time-series data, it’s difficult for comparative analysis of the water supply and demand situations in the past and present. Multi-period data can be accumulated in the future, and the water supply-demand balance model can be used for further analysis in future studies. Besides, the impacts of climate change on the water supply and demand situations can be assessed.

## 5. Conclusions

The vulnerability of different components can be compared horizontally based on the water vulnerability model, and it can easily identify the most vulnerable region because the result can be combined into an overall index. However, the water supply-demand balance model was superior to the vulnerability model in identifying the region with the most severe mismatch between supply and demand. The empirical findings showed that the hill districts had the highest water vulnerability and Tarai had the lowest vulnerability. The mismatch between agricultural water supply and demand was much more serious than that of domestic water at the basin level. In the three agro-ecological regions, Tarai encountered the most severe shortages of domestic water, while the hilly areas had the greatest pressure on shortages of agricultural water. Based on the combination of the DI and AI, the LL group (Lalitpur, Mahottari, Makwanpur, and Solukhumbu) was found to be the most water-stressed area due to shortages of both domestic and agricultural water, and funds for the construction of water supply systems should be given priority.

The improvement of water supply systems in the most water-stressed region is considered to be a priority, which is favorable to designing water supply systems that match the heterogeneous demands. In addition, adaptation strategies for mitigating water stress were proposed. The government should take charge of policy formulation and implementation and facilitate coordination and supervision among relevant departments, local communities, and households during the entire process of implementation. For example, water resources management requires relevant sectors to improve the coverage of water supply systems in water-stressed areas and promote the sustainable use of groundwater for irrigation. Local communities are the link between the government and inhabitants, and they can work with NGOs to build water tanks and provide technologies and services for residents to cope with droughts and reduce crop yield losses. In addition, households need to take multiple strategies to mitigate the impact of water shortages on their livelihoods. For instance, the water storage system is conducive to collecting rainwater. Water pipes and water channels can be used to transmit spring water from the sources to houses or farmland. Livelihood diversification is crucial to reducing water demand and water vulnerability. Minimizing the gap between water supply and demand to mitigate water stress requires the coordination and cooperation of multiple stakeholders, including the government, communities, and households.

## Figures and Tables

**Figure 1 ijerph-19-01606-f001:**
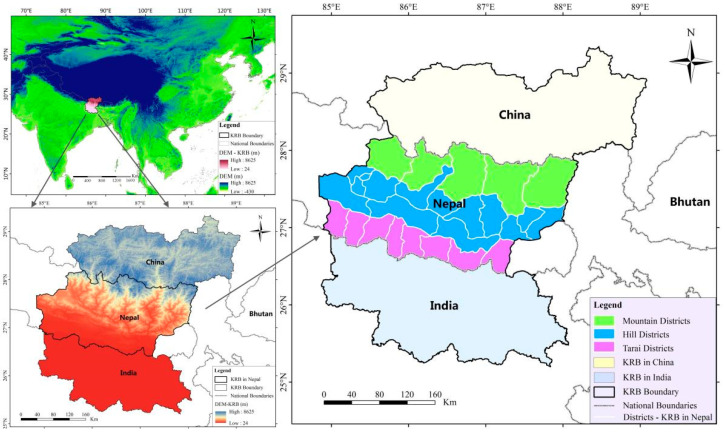
Location of the Koshi River basin.

**Figure 2 ijerph-19-01606-f002:**
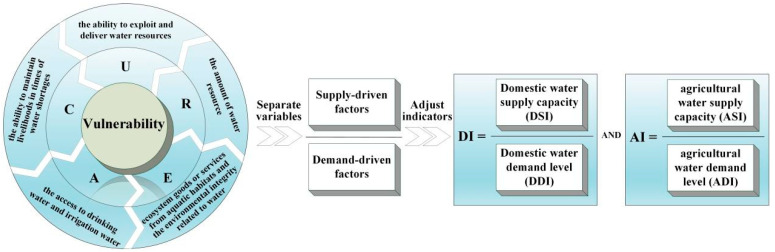
The construction process of frameworks for the water vulnerability and supply-demand balance index.

**Figure 3 ijerph-19-01606-f003:**
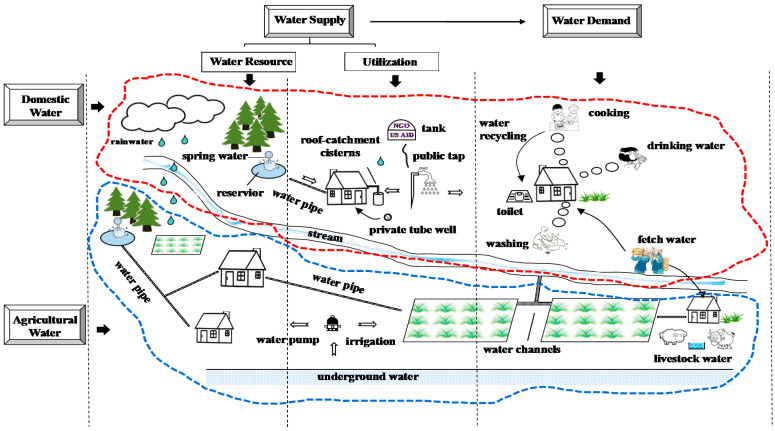
Water supply and demand situations in the KRB of Nepal.

**Figure 4 ijerph-19-01606-f004:**
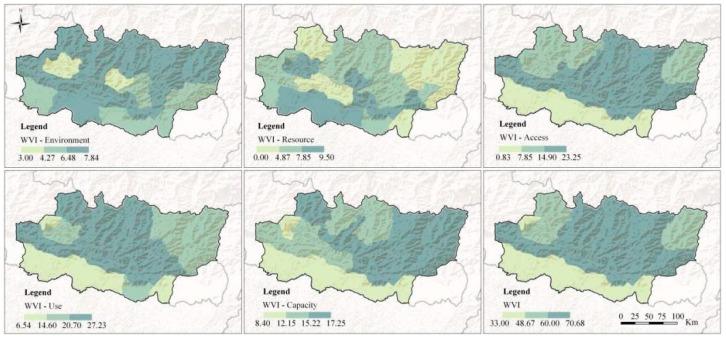
The spatial distribution of the WVI and the component vulnerability.

**Figure 5 ijerph-19-01606-f005:**
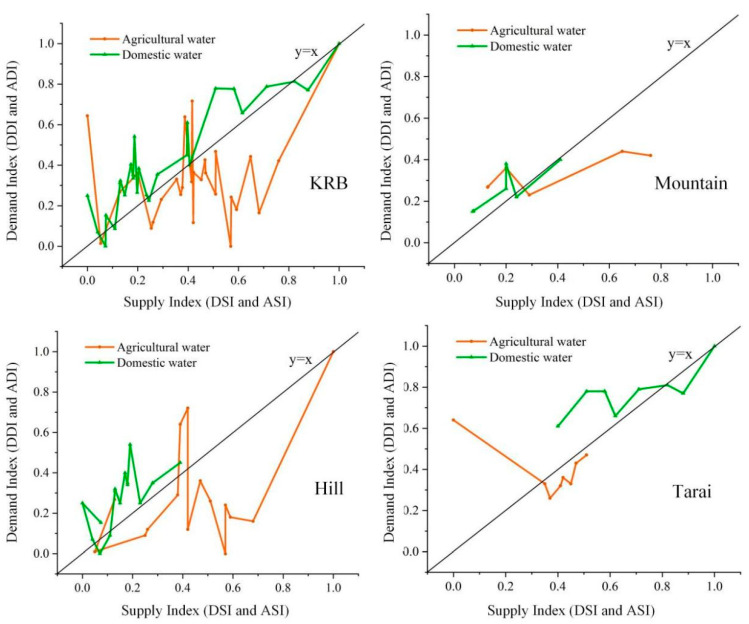
The DSI, DDI, ASI, and ADI in 27 districts and three agro-ecological regions.

**Figure 6 ijerph-19-01606-f006:**
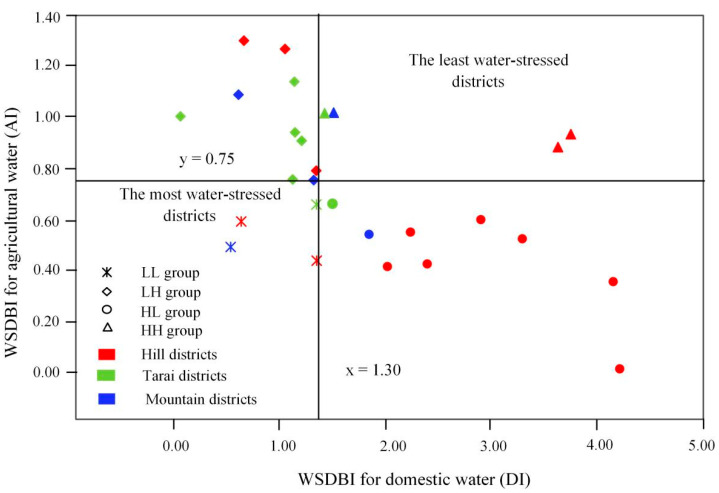
Classifications based on comparison with the median value.

**Figure 7 ijerph-19-01606-f007:**
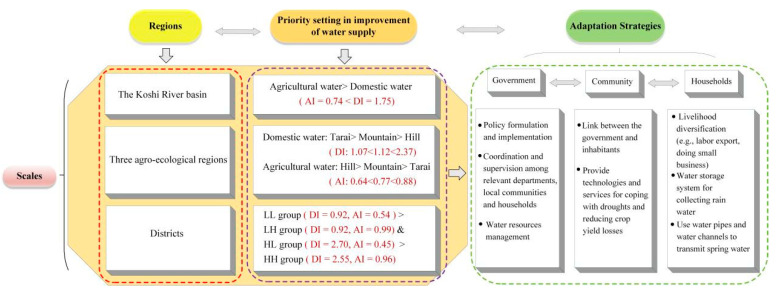
The priority setting in the improvement of water supply systems and adaptation strategies for mitigating water stress.

**Table 1 ijerph-19-01606-t001:** Primary variables used in the water vulnerability model and water supply-demand balance model.

Index	Dimension	Indicator	Anticipated Impact	References	Data Sources
WVI	E	pond area/m2 (per capita)	−	[41]	National Sample Census of Agriculture Nepal
		forest coverage (%)	−	[41]	National Sample Census of Agriculture Nepal
	R	annual precipitation (mm)	−	[21,46]	Global Climate Data
		coefficient of variation of precipitation	+	[12]	Global Climate Data
	A	households with access to source of irrigation (%)	−	[12,41]	National Sample Census of Agriculture Nepal
		households with access to drinking water (%)	−	[12,41]	Central Bureau of Statistics
	U	water sanitation coverage (%)	−	[52]	District Coverage of Water Supply and Sanitation
		percentage of irrigated to arable land (%)	−	[42]	Ministry Of Urban Development/Government
		water conservation awareness (%)	−	[42]	National Sample Census of Agriculture Nepal
	C	percentage of farm population (%)	+	[12,18]	National Sample Census of Agriculture Nepal
		per capita income (dollars)	−	[18,42]	Human Development Report
		households with irrigation infrastructure (%)	−	[53]	National Sample Census of Agriculture Nepal
		percentage of paddy field to arable land (%)	+	[12,18]	Statistical Year Book Of Nepal
		total number of livestock (number)	+	[12,18]	Ministry Of Urban Development/Government
WSDBI	DSI	annual precipitation (mm)	+	[21,46]	Department of Hydrology and Meteorology Nepal (DHM)
		coefficient of variation of precipitation	−	[12]	Department of Hydrology and Meteorology Nepal (DHM)
		annual water resources (m3/year)	+	[16]	[51,54]
		water supply coverage (%)	+	[50]	[55]
		percentage of whole-year water supply piped schemes (%)	+	[50]	[55]
	DDI	total population (number)	+	[18]	Statistical Year Book of Nepal-2015
		urbanization (%)	+	[45,56]	National Sample Census of Agriculture Nepal 2011/12
	ASI	annual precipitation (mm)	+	[20,46]	Department of Hydrology and Meteorology Nepal (DHM)
		coefficient of variation of precipitation	−	[12]	Department of Hydrology and Meteorology Nepal (DHM)
		annual water resources (m3/year)	+	[16]	[51,54]
		percentage of area equipped for irrigation (%)	+	[51,54]	[55]
		distance to water source (km)	+	[49]	Central Bureau of Statistics (CBS)-Nepal
	ADI	area of dry land (ha)	+	[48]	Statistical Year Book of Nepal-2015
		area of paddy field (ha)	+	[48]	Statistical Year Book of Nepal-2015
		total number of livestock (number)	+	[18]	[55]

**Table 2 ijerph-19-01606-t002:** The DI and AI in 27 districts.

Agro-Ecological Regions	Districts	DI	AI
Mountain	Sindhupalchok	1.80	0.53
	Sankhuwasabha	1.46	1.01
	Dolakha	1.27	0.75
	Solukhumbu	0.48	0.48
	Taplejung	0.55	1.09
Hill	Bhojpur	2.20	0.54
	Dhankuta	2.87	0.59
	Kavrepalanchok	4.14	0.35
	Khotang	3.27	0.52
	Bhaktapur	0.61	1.30
	Kathmandu	1.00	1.27
	Lalitpur	0.58	0.58
	Makwanpur	1.30	0.43
	Okhaldhunga	4.20	0.00
	Panchthar	1.98	0.41
	Ramechhap	2.36	0.42
	Sindhuli	1.30	0.78
	Terhathum	3.73	0.93
	Udayapur	3.61	0.88
Tarai	Bara	1.09	1.14
	Sunsari	0.00	1.00
	Dhanusa	1.16	0.90
	Mahottari	1.30	0.65
	Rautahat	1.10	0.94
	Saptari	1.07	0.75
	Sarlahi	1.38	1.01
	Siraha	1.45	0.65

## Data Availability

The datasets generated during the current study are available from the corresponding author on reasonable request.

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
