# Peer review of "Application of a Water Supply-Demand Balance Model to Set Priorities for Improvements in Water Supply Systems: A Case Study from the Koshi River Basin, Nepal"

_ijerph, 2022, doi:10.3390/ijerph19031606_

Round 1
Reviewer 1 Report
In their paper, Application of a water supply-demand balance model to set priorities for improvements in water supply systems: a case study from the Koshi River Basin, Nepal, Zhu and Fang develop and apply methods for improving upon previous water vulnerability models to inform water improvement priorities in a targeted region. Their approach, which is well considered, relies on models that compare water demand for domestic and agricultural purposes with available supplies in different geographies and districts of the Koshi River Basin. Their results are not surprising: domestic water access presents a greater challenge in the plains of the Tarai due to population growth and water for agriculture is more limited in hilly areas due to the topography. However, the study also ranked districts according to their water stress to identify priority areas for water supply interventions.
Comments:
- Figures in scientific papers must include legends that allow readers to fully understand the material presented in the figures without referencing the text. This is not the case for Figures 4, 5, and 6 in the manuscript. The authors must clearly specify the significance of each figure, the findings that are illustrated, provide guidance on understanding the figures, and spell-out acronyms. For example: I do not understand how the graphs shown in Figure 5 are related to specific geographic locations.
- Similarly, what does "Relation with index" mean in Table 1?
- A table that presents the 27 study districts according their combination of DI and AI will improve the presentation.
- This study is largely based on data collected by the Govt. of Nepal. The authors must provide some background on data quality: are the datasets up to date? Did they crosscheck the data? Are the data reliable?
- Related to the data issue, the authors must describe the limitations of this research and options for building upon and improving the study.
- It will be helpful for the authors to comment on how climate change may affect (or is already affecting) their findings.
Author Response
Reviewer1:
- Figures in scientific papers must include legends that allow readers to fully understand the material presented in the figures without referencing the text. This is not the case for Figures 4, 5, and 6 in the manuscript. The authors must clearly specify the significance of each figure, the findings that are illustrated, provide guidance on understanding the figures, and spell-out acronyms. For example: I do not understand how the graphs shown in Figure 5 are related to specific geographic locations.
[Reply to the comment]
We appreciate the reviewer’s comments. We have supplemented the legend and illustrated which component of the water vulnerability index each graph represented in Figure 4. The first graph shown in Figure 5 is based on the DSI, DDI, ASI and ADI in 27 districts, and the 2-4 graphs are based on the DSI, DDI, ASI and ADI in districts which belong to corresponding geographic locations. Colors represent different geographical locations, and graphic shapes represent different group classes in Figure 6, which is shown in the legend.
- Similarly, what does "Relation with index" mean in Table 1?
[Reply to the comment]
We appreciate the reviewer’s comments. We have substituted “Anticipated impact” for “Relation with index”. It means whether this indicator is positive or negative.
- A table that presents the 27 study districts according their combination of DI and AI will improve the presentation.
[Reply to the comment]
We appreciate the reviewer’s comments. We have supplemented a table (Table 2) that presented the 27 study districts according their combination of DI and AI.
- This study is largely based on data collected by the Govt. of Nepal. The authors must provide some background on data quality: are the datasets up to date? Did they crosscheck the data? Are the data reliable?
[Reply to the comment]
We appreciate the reviewer’s comments. Most of Nepal's socio-economic data are not updated annually. The data used in this paper are all from the latest datasets published by various government departments of Nepal (e.g., ICIMOD, Central Bureau of Statistics and Department of Hydrology and Meteorology Nepal). They are authoritative and have been used in many research. We have supplemented some background on data quality in data sources section.
- Related to the data issue, the authors must describe the limitations of this research and options for building upon and improving the study. 6. It will be helpful for the authors to comment on how climate change may affect (or is already affecting) their findings.
[Reply to the comment]
We appreciate the reviewer’s comments. Related to the data issue, we have supplemented “4.4. Limitations of this study” section as follows:
Due to the lack of time series data, it’s difficult for comparative analysis of the water supply and demand situations in the past and present. Multi-period data can be accumulated in the future, and the water supply-demand balance model can be used for further analysis in future studies. Besides, the impacts of climate change on the water supply and demand situations can be assessed.

Reviewer 2 Report
The paper deals with the proposal of a water supply-demand balance model to quantify the supply –demand gap. The authors describe an analysis based on three steps: to construct the balance model, to identify the region with most severe mismatch, to illustrate the priority setting for the improvement of water supply systems and to adapt strategies.
The paper is well constructed and both the introduction and the other parts of the work are well proposed.
The authors should clarify only few aspects:
- it is important to define better how to obtain the weights of the five component for WVI , they values and to explain how the results can change when they assume different values;
The same suggestion is for the others coefficient: there is no indication to the range of values they can assume.
When the authors define the priority they don’t give any indication about the cost to obtain the requested results. Can they give some information about this topic and about the time necessary to achieve the goals? In other term it is necessary a brief analysis of the costs of the proposed approach
Author Response
Reviewer2:
- it is important to define better how to obtain the weights of the five component for WVI , they values and to explain how the results can change when they assume different values;The same suggestion is for the others coefficient: there is no indication to the range of values they can assume.
[Reply to the comment]
We appreciate the reviewer’s comments. We adopted the balanced weighted average approach to model the vulnerability to water shortages based on the assumption that each component equally contributes to the WVI, as mentioned in 2.4.1. Besides, we supplemented information about the range of values the component can assume and how the results can change as follows:
The values of five components all range from 0 to 0.2, and the higher the component value, the higher the WVI.
The DSI, DDI, ASI and ADI all range from 0 to 1. The higher DSI/ASI and lower DDI/ADI, the higher DI/AI. If DI/AI is less than 1, supply is less than demand; if DI/AI is equal to 1, water supply and demand are in balance; if DI/AI is more than 1, demand is less than supply.
- When the authors define the priority they don’t give any indication about the cost to obtain the requested results. Can they give some information about this topic and about the time necessary to achieve the goals? In other term it is necessary a brief analysis of the costs of the proposed approach
[Reply to the comment]
We appreciate the reviewer’s comments. The model proposed in this study highlights the region with the most severe mismatch between water supply and demand, which helps to attract the attention of the government, nongovernmental organizations (NGOs) and communities, and provides a reference for them to set priorities for the improvement of water supply systems. However, the costs and time of the model are not covered in our study.
